# Regularization-Free Estimation in Trace Regression with Symmetric Positive Semidefinite Matrices

**Martin Slawski**          **Ping Li**
Department of Statistics & Biostatistics
Department of Computer Science
Rutgers University
Piscataway, NJ 08854, USA
{martin.slawski@rutgers.edu,
  pingli@stat.rutgers.edu}

**Matthias Hein**
Department of Computer Science
Department of Mathematics
Saarland University
Saarbrücken, Germany
hein@cs.uni-saarland.de

## Abstract

Trace regression models have received considerable attention in the context of matrix completion, quantum state tomography, and compressed sensing. Estimation of the underlying matrix from regularization-based approaches promoting low-rankedness, notably nuclear norm regularization, have enjoyed great popularity. In this paper, we argue that such regularization may no longer be necessary if the underlying matrix is symmetric positive semidefinite (spd) and the design satisfies certain conditions. In this situation, simple least squares estimation subject to an spd constraint may perform as well as regularization-based approaches with a proper choice of regularization parameter, which entails knowledge of the noise level and/or tuning. By contrast, constrained least squares estimation comes without any tuning parameter and may hence be preferred due to its simplicity.

## 1 Introduction

Trace regression models of the form

$$y_i = \operatorname{tr}(X_i^\top \Sigma^*) + \varepsilon_i, \quad i = 1, \ldots, n, \tag{1}$$

where $\Sigma^* \in \mathbb{R}^{m_1 \times m_2}$ is the parameter of interest to be estimated given measurement matrices $X_i \in \mathbb{R}^{m_1 \times m_2}$ and observations $y_i$ contaminated by errors $\varepsilon_i$, $i = 1, \ldots, n$, have attracted considerable interest in high-dimensional statistical inference, machine learning and signal processing over the past few years. Research in these areas has focused on a setting with few measurements $n \ll m_1 \cdot m_2$ and $\Sigma^*$ being (approximately) of low rank $r \ll \min\{m_1, m_2\}$. Such setting is relevant to problems such as matrix completion [6, 23], compressed sensing [5, 17], quantum state tomography [11] and phase retrieval [7]. A common thread in these works is the use of the nuclear norm of a matrix as a convex surrogate for its rank [18] in regularized estimation amenable to modern optimization techniques. This approach can be seen as natural generalization of $\ell_1$-norm (aka lasso) regularization for the linear regression model [24] that arises as a special case of model (1) in which both $\Sigma^*$ and $\{X_i\}_{i=1}^n$ are diagonal. It is inarguable that in general regularization is essential if $n < m_1 \cdot m_2$.

The situation is less clear if $\Sigma^*$ is known to satisfy additional constraints that can be incorporated in estimation. Specifically, in the present paper we consider the case in which $m_1 = m_2 = m$ and $\Sigma^*$ is known to be symmetric positive semidefinite (spd), i.e. $\Sigma^* \in \mathbb{S}_+^m$ with $\mathbb{S}_+^m$ denoting the positive semidefinite cone in the space of symmetric real $m \times m$ matrices $\mathbb{S}^m$. The set $\mathbb{S}_+^m$ deserves specific interest as it includes covariance matrices and Gram matrices in kernel-based learning [20]. It is rather common for these matrices to be of low rank (at least approximately), given the widespread use of principal components analysis and low-rank kernel approximations [28]. In the present paper, we focus on the usefulness of the spd constraint for estimation. We argue that if $\Sigma^*$ is spd and the measurement matrices $\{X_i\}_{i=1}^n$ obey certain conditions, constrained least squares estimation

$$\min_{\Sigma \in \mathbb{S}_+^m} \frac{1}{2n} \sum_{i=1}^n (y_i - \operatorname{tr}(X_i^\top \Sigma))^2 \tag{2}$$

may perform similarly well in prediction and parameter estimation as approaches employing nuclear norm regularization with proper choice of the regularization parameter, including the interesting

regime $n < \delta_m$, where $\delta_m = \dim(\mathbb{S}^m) = m(m+1)/2$. Note that the objective in (2) only consists of a data fitting term and is hence convenient to work with in practice since there is no free parameter. Our findings can be seen as a non-commutative extension of recent results on non-negative least squares estimation for linear regression [16, 21].

**Related work.** Model (1) with $\Sigma^* \in \mathbb{S}_+^m$ has been studied in several recent papers. A good deal of these papers consider the setup of compressed sensing in which the $\{X_i\}_{i=1}^n$ can be chosen by the user, with the goal to minimize the number of observations required to (approximately) recover $\Sigma^*$. For example, in [27], recovery of $\Sigma^*$ being low-rank from noiseless observations ($\varepsilon_i = 0$, $i = 1, \ldots, n$) by solving a feasibility problem over $\mathbb{S}_+^m$ is considered, which is equivalent to the constrained least squares problem (1) in a noiseless setting.

In [3, 8], recovery from rank-one measurements is considered, i.e., for $\{x_i\}_{i=1}^n \subset \mathbb{R}^m$

$$y_i = x_i^\top \Sigma^* x_i + \varepsilon_i = \mathrm{tr}(X_i^\top \Sigma^*) + \varepsilon_i, \quad \text{with } X_i = x_i x_i^\top, \ i = 1, \ldots, n. \tag{3}$$

As opposed to [3, 8], where estimation based on nuclear norm regularization is proposed, the present work is devoted to regularization-free estimation. While rank-one measurements as in (3) are also in the center of interest herein, our framework is not limited to this case. In [3] an application of (3) to covariance matrix estimation given only one-dimensional projections $\{x_i^\top z_i\}_{i=1}^n$ of the data points is discussed, where the $\{z_i\}_{i=1}^n$ are i.i.d. from a distribution with zero mean and covariance matrix $\Sigma^*$. In fact, this fits the model under study with observations

$$y_i = (x_i^\top z_i)^2 = x_i^\top z_i z_i^\top x_i = x_i^\top \Sigma^* x_i + \varepsilon_i, \quad \varepsilon_i = x_i^\top \{z_i z_i^\top - \Sigma^*\} x_i, \ i = 1, \ldots, n. \tag{4}$$

Specializing (3) to the case in which $\Sigma^* = \sigma^*(\sigma^*)^\top$, one obtains the quadratic model

$$y_i = |x_i^\top \sigma^*|^2 + \varepsilon_i \tag{5}$$

which (with complex-valued $\sigma^*$) is relevant to the problem of phase retrieval [14]. The approach of [7] treats (5) as an instance of (1) and uses nuclear norm regularization to enforce rank-one solutions. In follow-up work [4], the authors show a refined recovery result stating that imposing an spd constraint − without regularization − suffices. A similar result has been proven independently by [10]. However, the results in [4] and [10] only concern model (5). After posting an extended version [22] of the present paper, a generalization of [4, 10] to general low-rank spd matrices has been achieved in [13]. Since [4, 10, 13] consider bounded noise, whereas the analysis herein assumes Gaussian noise, our results are not direclty comparable to those in [4, 10, 13].

**Notation.** $\mathbb{M}^d$ denotes the space of real $d \times d$ matrices with inner product $\langle M, M' \rangle := \mathrm{tr}(M^\top M')$. The subspace of symmetric matrices $\mathbb{S}^d$ has dimension $\delta_d := d(d+1)/2$. $M \in \mathbb{S}^d$ has an eigen-decomposition $M = U\Lambda U^\top = \sum_{j=1}^d \lambda_j(M) u_j u_j^\top$, where $\lambda_1(M) = \lambda_{\max}(M) \geq \lambda_2(M) \geq \ldots \geq \lambda_d(M) = \lambda_{\min}(M)$, $\Lambda = \mathrm{diag}(\lambda_1(M), \ldots, \lambda_d(M))$, and $U = [u_1 \ldots u_d]$. For $q \in [1, \infty)$ and $M \in \mathbb{S}^d$, $\|M\|_q := (\sum_{j=1}^d |\lambda_j(M)|^q)^{1/q}$ denotes the Schatten-$q$-norm ($q = 1$: nuclear norm; $q = 2$ Frobenius norm $\|M\|_F$, $q = \infty$: spectral norm $\|M\|_\infty := \max_{1 \leq j \leq d} |\lambda_j(M)|$). Let $\mathcal{S}_1(d) = \{M \in \mathbb{S}^d : \|M\|_1 = 1\}$ and $\mathcal{S}_1^+(d) = \mathcal{S}_1(d) \cap \mathbb{S}_+^d$. The symbols $\succeq, \preceq, \succ, \prec$ refer to the semidefinite ordering. For a set $A$ and $\alpha \in \mathbb{R}$, $\alpha A := \{\alpha a, a \in A\}$.

It is convenient to re-write model (1) as $y = \mathcal{X}(\Sigma^*) + \varepsilon$, where $y = (y_i)_{i=1}^n$, $\varepsilon = (\varepsilon_i)_{i=1}^n$ and $\mathcal{X} : \mathbb{M}^m \to \mathbb{R}^n$ is a linear map defined by $(\mathcal{X}(M))_i = \mathrm{tr}(X_i^\top M)$, $i = 1, \ldots, n$, referred to as *sampling operator*. Its adjoint $\mathcal{X}^* : \mathbb{R}^n \to \mathbb{M}^m$ is given by the map $v \mapsto \sum_{i=1}^n v_i X_i$.

**Supplement.** The appendix contains all proofs, additional experiments and figures.

## 2 Analysis

**Preliminaries.** Throughout this section, we consider a special instance of model (1) in which

$$y_i = \mathrm{tr}(X_i \Sigma^*) + \varepsilon_i, \quad \text{where } \Sigma^* \in \mathbb{S}_+^m, \ X_i \in \mathbb{S}^m, \text{ and } \varepsilon_i \overset{\text{i.i.d.}}{\sim} N(0, \sigma^2), \ i = 1, \ldots, n. \tag{6}$$

The assumption that the errors $\{\varepsilon_i\}_{i=1}^n$ are Gaussian is made for convenience as it simplifies the stochastic part of our analysis, which could be extended to sub-Gaussian errors.

Note that w.l.o.g., we may assume that $\{X_i\}_{i=1}^n \subset \mathbb{S}^m$. In fact, since $\Sigma^* \in \mathbb{S}^m$, for any $M \in \mathbb{M}^m$ we have that $\mathrm{tr}(M\Sigma^*) = \mathrm{tr}(M^{\mathrm{sym}}\Sigma^*)$, where $M^{\mathrm{sym}} = (M + M^\top)/2$.

In the sequel, we study the statistical performance of the constrained least squares estimator

$$\widehat{\Sigma} \in \underset{\Sigma \in \mathbb{S}_+^m}{\mathrm{argmin}} \frac{1}{2n} \|y - \mathcal{X}(\Sigma)\|_2^2 \qquad (7)$$

under model (6). More specifically, under certain conditions on $\mathcal{X}$, we shall derive bounds on

$$(a) \quad \frac{1}{n} \|\mathcal{X}(\Sigma^*) - \mathcal{X}(\widehat{\Sigma})\|_2^2, \quad \text{and} \quad (b) \quad \|\widehat{\Sigma} - \Sigma^*\|_1, \qquad (8)$$

where $(a)$ will be referred to as "prediction error" below. The most basic method for estimating $\Sigma^*$ is ordinary least squares (ols) estimation

$$\widehat{\Sigma}^{\mathrm{ols}} \in \underset{\Sigma \in \mathbb{S}^m}{\mathrm{argmin}} \frac{1}{2n} \|y - \mathcal{X}(\Sigma)\|_2^2, \qquad (9)$$

which is computationally simpler than (7). While (7) requires convex programming, (9) boils down to solving a linear system of equations in $\delta_m = m(m+1)/2$ variables. On the other hand, the prediction error of ols scales as $O_{\mathbf{P}}(\dim(\mathrm{range}(\mathcal{X}))/n)$, where $\dim(\mathrm{range}(\mathcal{X}))$ can be as large as $\min\{n, \delta_m\}$, in which case the prediction error vanishes only if $\delta_m/n \to 0$ as $n \to \infty$. Moreover, the estimation error $\|\widehat{\Sigma}^{\mathrm{ols}} - \Sigma^*\|_1$ is unbounded unless $n \geq \delta_m$. Research conducted over the past few years has thus focused on methods dealing successfully with the case $n < \delta_m$ as long as the target $\Sigma^*$ has additional structure, notably low-rankedness. Indeed, if $\Sigma^*$ has rank $r \ll m$, the intrinsic dimension of the problem becomes (roughly) $mr \ll \delta_m$. In a large body of work, nuclear norm regularization, which serves as a convex surrogate of rank regularization, is considered as a computationally convenient alternative for which a series of adaptivity properties to underlying low-rankedness has been established, e.g. [5, 15, 17, 18, 19]. Complementing (9) with nuclear norm regularization yields the estimator

$$\widehat{\Sigma}^1 \in \underset{\Sigma \in \mathbb{S}^m}{\mathrm{argmin}} \frac{1}{2n} \|y - \mathcal{X}(\Sigma)\|_2^2 + \lambda \|\Sigma\|_1, \qquad (10)$$

where $\lambda > 0$ is a regularization parameter. In case an spd constraint is imposed (10) becomes

$$\widehat{\Sigma}^{1+} \in \underset{\Sigma \in \mathbb{S}_+^m}{\mathrm{argmin}} \frac{1}{2n} \|y - \mathcal{X}(\Sigma)\|_2^2 + \lambda \mathrm{tr}(\Sigma). \qquad (11)$$

Our analysis aims at elucidating potential advantages of the spd constraint in the constrained least squares problem (7) from a statistical point of view. It turns out that depending on properties of $\mathcal{X}$, the behaviour of $\widehat{\Sigma}$ can range from a performance similar to the least squares estimator $\widehat{\Sigma}^{\mathrm{ols}}$ on the one hand to a performance similar to the nuclear norm regularized estimator $\widehat{\Sigma}^{1+}$ with properly chosen/tuned $\lambda$ on the other hand. The latter case appears to be remarkable: $\widehat{\Sigma}$ may enjoy similar adaptivity properties as nuclear norm regularized estimators even though $\widehat{\Sigma}$ is obtained from a pure data fitting problem without explicit regularization.

## 2.1 Negative result

We first discuss a negative example of $\mathcal{X}$ for which the spd-constrained estimator $\widehat{\Sigma}$ does not improve (substantially) over the unconstrained estimator $\widehat{\Sigma}^{\mathrm{ols}}$. At the same time, this example provides clues on conditions to be imposed on $\mathcal{X}$ to achieve substantially better performance.

**Random Gaussian design.** Consider the Gaussian orthogonal ensemble (GOE)

$$\mathrm{GOE}(m) = \{X = (x_{jk}), \ \{x_{jj}\}_{j=1}^m \overset{\mathrm{i.i.d.}}{\sim} N(0,1), \ \{x_{jk} = x_{kj}\}_{1 \leq j < k \leq m} \overset{\mathrm{i.i.d.}}{\sim} N(0, 1/2)\}.$$

Gaussian measurements are common in compressed sensing. It is hence of interest to study measurements $\{X_i\}_{i=1}^n \overset{\mathrm{i.i.d.}}{\sim} \mathrm{GOE}(m)$ in the context of the constrained least squares problem (7). The following statement points to a serious limitation associated with such measurements.

**Proposition 1.** *Consider $X_i \overset{i.i.d.}{\sim} GOE(m)$, $i = 1, \ldots, n$. For any $\varepsilon > 0$, if $n \leq (1 - \varepsilon)\delta_m/2$, with probability at least $1 - 32\exp(-\varepsilon^2\delta_m)$, there exists $\Delta \in \mathbb{S}_+^m$, $\Delta \neq 0$ such that $\mathcal{X}(\Delta) = 0$.*

Proposition 1 implies that if the number of measurements drops below $1/2$ of the ambient dimension $\delta_m$, estimating $\Sigma^*$ based on (7) becomes ill-posed; the estimation error $\|\widehat{\Sigma} - \Sigma^*\|_1$ is unbounded, irrespective of the rank of $\Sigma^*$. Geometrically, the consequence of Proposition 1 is that the convex cone $\mathcal{C}_{\mathcal{X}} = \{z \in \mathbb{R}^n : z = \mathcal{X}(\Delta), \ \Delta \in \mathbb{S}_+^m\}$ contains 0. Unless 0 is contained in the boundary of $\mathcal{C}_{\mathcal{X}}$ (we conjecture that this event has measure zero), this means that $\mathcal{C}_{\mathcal{X}} = \mathbb{R}^n$, i.e. the spd constraint becomes vacuous.

## 2.2 Slow Rate Bound on the Prediction Error

We present a positive result on the $\mathsf{spd}$-constrained least squares estimator $\widehat{\Sigma}$ under an additional condition on the sampling operator $\mathcal{X}$. Specifically, the prediction error will be bounded as

$$\frac{1}{n}\|\mathcal{X}(\Sigma^*) - \mathcal{X}(\widehat{\Sigma})\|_2^2 = O(\lambda_0\|\Sigma^*\|_1 + \lambda_0^2), \quad \text{where } \lambda_0 = \frac{1}{n}\|\mathcal{X}^*(\varepsilon)\|_\infty, \qquad (12)$$

with $\lambda_0$ typically being of the order $O(\sqrt{m/n})$ (up to log factors). The rate in (12) can be a significant improvement of what is achieved by $\widehat{\Sigma}^{\mathrm{ols}}$ if $\|\Sigma^*\|_1 = \mathrm{tr}(\Sigma^*)$ is small. If $\lambda_0 = o(\|\Sigma^*\|_1)$ that rate coincides with those of the nuclear norm regularized estimators (10), (11) with regularization parameter $\lambda \geq \lambda_0$, cf. Theorem 1 in [19]. For nuclear norm regularized estimators, the rate $O(\lambda_0\|\Sigma^*\|_1)$ is achieved for any choice of $\mathcal{X}$ and is slow in the sense that the squared prediction error only decays at the rate $n^{-1/2}$ instead of $n^{-1}$.

**Condition on $\mathcal{X}$.** In order to arrive at a suitable condition to be imposed on $\mathcal{X}$ so that (12) can be achieved, it makes sense to re-consider the negative example of Proposition 1, which states that as long as $n$ is bounded away from $\delta_m/2$ from above, there is a non-trivial $\Delta \in \mathbb{S}_+^m$ such that $\mathcal{X}(\Delta) = 0$. Equivalently, $\mathrm{dist}(\mathcal{P}_\mathcal{X}, 0) = \min_{\Delta \in \mathcal{S}_1^+(m)}\|\mathcal{X}(\Delta)\|_2 = 0$, where

$$\mathcal{P}_\mathcal{X} := \{z \in \mathbb{R}^n : z = \mathcal{X}(\Delta), \Delta \in \mathcal{S}_1^+(m)\}, \quad \text{and } \mathcal{S}_1^+(m) := \{\Delta \in \mathbb{S}_+^m : \mathrm{tr}(\Delta) = 1\}.$$

In this situation, it is impossible to derive a non-trivial bound on the prediction error as $\mathrm{dist}(\mathcal{P}_\mathcal{X}, 0) = 0$ may imply $\mathcal{C}_\mathcal{X} = \mathbb{R}^n$ so that $\|\mathcal{X}(\Sigma^*) - \mathcal{X}(\widehat{\Sigma})\|_2^2 = \|\varepsilon\|_2^2$. To rule this out, the condition $\mathrm{dist}(\mathcal{P}_\mathcal{X}, 0) > 0$ is natural. More strongly, one may ask for the following:

$$\text{There exists a constant } \tau > 0 \text{ such that } \tau_0^2(\mathcal{X}) := \min_{\Delta \in \mathcal{S}_1^+(m)} \frac{1}{n}\|\mathcal{X}(\Delta)\|_2^2 \geq \tau^2. \qquad (13)$$

An analogous condition is sufficient for a slow rate bound in the vector case, cf. [21]. However, the condition for the slow rate bound in Theorem 1 below is somewhat stronger than (13).

**Condition 1.** *There exist constants $R_* > 1$, $\tau_* > 0$ s.t. $\tau^2(\mathcal{X}, R_*) \geq \tau_*^2$, where for $R \in \mathbb{R}$*

$$\tau^2(\mathcal{X}, R) = \mathrm{dist}^2(R\mathcal{P}_\mathcal{X}, \mathcal{P}_\mathcal{X})/n = \min_{\substack{A \in R\mathcal{S}_1^+(m) \\ B \in \mathcal{S}_1^+(m)}} \frac{1}{n}\|\mathcal{X}(A) - \mathcal{X}(B)\|_2^2.$$

The following condition is sufficient for Condition 1 and in some cases much easier to check.

**Proposition 2.** *Suppose there exists $a \in \mathbb{R}^n$, $\|a\|_2 \leq 1$, and constants $0 < \phi_{\min} \leq \phi_{\max}$ s.t.*

$$\lambda_{\min}(n^{-1/2}\mathcal{X}^*(a)) \geq \phi_{\min}, \quad \text{and } \lambda_{\max}(n^{-1/2}\mathcal{X}^*(a)) \leq \phi_{\max}.$$

*Then for any $\zeta > 1$, $\mathcal{X}$ satisfies Condition 1 with $R_* = \zeta(\phi_{\max}/\phi_{\min})$ and $\tau_*^2 = (\zeta - 1)^2\phi_{\max}^2$.*

The condition of Proposition 2 can be phrased as having a positive definite matrix in the image of the unit ball under $\mathcal{X}^*$, which, after scaling by $1/\sqrt{n}$, has its smallest eigenvalue bounded away from zero and a bounded condition number. As a simple example, suppose that $X_1 = \sqrt{n}I$. Invoking Proposition 2 with $a = (1, 0, \ldots, 0)^\top$ and $\zeta = 2$, we find that Condition 1 is satisfied with $R_* = 2$ and $\tau_*^2 = 1$. A more interesting example is random design where the $\{X_i\}_{i=1}^n$ are (sample) covariance matrices, where the underlying random vectors satisfy appropriate tail or moment conditions.

**Corollary 1.** *Let $\pi_m$ be a probability distribution on $\mathbb{R}^m$ with second moment matrix $\Gamma := \mathbf{E}_{z \sim \pi_m}[zz^\top]$ satisfying $\lambda_{\min}(\Gamma) > 0$. Consider the random matrix ensemble*

$$\mathcal{M}(\pi_m, q) = \left\{\frac{1}{q}\sum_{k=1}^q z_k z_k^\top, \{z_k\}_{k=1}^q \overset{i.i.d.}{\sim} \pi_m\right\}. \qquad (14)$$

*Suppose that $\{X_i\}_{i=1}^n \overset{i.i.d.}{\sim} \mathcal{M}(\pi_m, q)$ and let $\widehat{\Gamma}_n := \frac{1}{n}\sum_{i=1}^n X_i$ and $0 < \epsilon_n < \lambda_{\min}(\Gamma)$. Under the event $\{\|\Gamma - \widehat{\Gamma}_n\|_\infty \leq \epsilon_n\}$, $\mathcal{X}$ satisfies Condition 1 with*

$$R_* = \frac{2(\lambda_{\max}(\Gamma) + \epsilon_n)}{\lambda_{\min}(\Gamma) - \epsilon_n} \quad \text{and} \quad \tau_*^2 = (\lambda_{\max}(\Gamma) + \epsilon_n)^2.$$

It is instructive to spell out Corollary 1 with $\pi_m$ as the standard Gaussian distribution on $\mathbb{R}^m$. The matrix $\widehat{\Gamma}_n$ equals the sample covariance matrix computed from $N = n \cdot q$ samples. It is well-known [9] that for $m, N$ large, $\lambda_{\max}(\widehat{\Gamma}_n)$ and $\lambda_{\min}(\widehat{\Gamma}_n)$ concentrate sharply around $(1+\eta_n)^2$ and $(1-\eta_n)^2$, respectively, where $\eta_n = \sqrt{m/N}$. Hence, for any $\gamma > 0$, there exists $C_\gamma > 1$ so that if $N \geq C_\gamma m$, it holds that $R_* \leq 2 + \gamma$. Similar though weaker concentration results for $\|\Gamma - \widehat{\Gamma}_n\|_\infty$ exist for the broader class of distributions $\pi_m$ with finite fourth moments [26]. Specialized to $q = 1$, Corollary 1 yields a statement about $\mathcal{X}$ made up from random rank-one measurements $X_i = z_i z_i^\top$, $i = 1, \ldots, n$, cf. (3). The preceding discussion indicates that Condition 1 tends to be satisfied in this case.

**Theorem 1.** *Suppose that model* (6) *holds with $\mathcal{X}$ satisfying Condition 1 with constants $R_*$ and $\tau_*^2$. We then have*

$$\frac{1}{n}\|\mathcal{X}(\Sigma^*) - \mathcal{X}(\widehat{\Sigma})\|_2^2 \leq \max\left\{ 2(1 + R_*)\lambda_0\|\Sigma^*\|_1, \ 2\lambda_0\|\Sigma^*\|_1 + 8\left(\lambda_0\frac{R_*}{\tau_*}\right)^2\right\}$$

*where, for any $\mu \geq 0$, with probability at least $1 - (2m)^{-\mu}$*

$$\lambda_0 \leq \sigma\sqrt{(1 + \mu)2\log(2m)\frac{V_n^2}{n}}, \quad \text{where} \quad V_n^2 = \left\|\frac{1}{n}\sum_{i=1}^n X_i^2\right\|_\infty.$$

**Remark:** Under the scalings $R_* = O(1)$ and $\tau_*^2 = \Omega(1)$, the bound of Theorem 1 is of the order $O(\lambda_0\|\Sigma^*\|_1 + \lambda_0^2)$ as announced at the beginning of this section. For given $\mathcal{X}$, the quantity $\tau^2(\mathcal{X}, R)$ can be evaluated by solving a least squares problem with spd constraints. Hence it is feasible to check in practice whether Condition 1 holds. For later reference, we evaluate the term $V_n^2$ for $\mathcal{M}(\pi_m, q)$ with $\pi_m$ as standard Gaussian distribution. As shown in the supplement, with high probability, $V_n^2 = O(m \log n)$ holds as long as $m = O(nq)$.

## 2.3 Bound on the Estimation Error

In the previous subsection, we did not make any assumptions about $\Sigma^*$ apart from $\Sigma^* \in \mathbb{S}_+^m$. Henceforth, we suppose that $\Sigma^*$ is of low rank $1 \leq r \ll m$ and study the performance of the constrained least squares estimator (7) for prediction and estimation in such setting.

**Preliminaries.** Let $\Sigma^* = U\Lambda U^\top$ be the eigenvalue decomposition of $\Sigma^*$, where

$$U = \left[\begin{array}{cc} U_\| & U_\perp \\ m \times r & m \times (m - r) \end{array}\right]\left[\begin{array}{cc} \Lambda_r & 0_{r\times(m-r)} \\ 0_{(m-r)\times r} & 0_{(m-r)\times(m-r)} \end{array}\right]$$

where $\Lambda_r$ is diagonal with positive diagonal entries. Consider the linear subspace

$$\mathbb{T}^\perp = \{M \in \mathbb{S}^m : \ M = U_\perp A U_\perp^\top, \quad A \in \mathbb{S}^{m-r}\}.$$

From $U_\perp^\top\Sigma^* U_\perp = 0$, it follows that $\Sigma^*$ is contained in the orthogonal complement

$$\mathbb{T} = \{M \in \mathbb{S}^m : \ M = U_\| B + B^\top U_\|^\top, \quad B \in \mathbb{R}^{r\times m}\},$$

of dimension $mr - r(r - 1)/2 \ll \delta_m$ if $r \ll m$. The image of $\mathbb{T}$ under $\mathcal{X}$ is denoted by $\mathcal{T}$.

**Conditions on $\mathcal{X}$.** We introduce the key quantities the bound in this subsection depends on.
*Separability constant.*

$$\tau^2(\mathbb{T}) = \frac{1}{n}\text{dist}^2(\mathcal{T}, \mathcal{P}_\mathcal{X}), \quad \mathcal{P}_\mathcal{X} := \{z \in \mathbb{R}^n : \ z = \mathcal{X}(\Delta), \ \Delta \in \mathbb{T}^\perp \cap \mathcal{S}_1^+(m)\}$$

$$= \min_{\Theta \in \mathbb{T}, \ \Lambda \in \mathcal{S}_1^+(m)\cap\mathbb{T}^\perp}\frac{1}{n}\|\mathcal{X}(\Theta) - \mathcal{X}(\Lambda)\|_2^2$$

*Restricted eigenvalue.*

$$\phi^2(\mathbb{T}) = \min_{0\neq\Delta\in\mathbb{T}}\frac{\|\mathcal{X}(\Delta)\|_2^2/n}{\|\Delta\|_1^2}.$$

As indicated by the following statement concerning the noiseless case, for bounding $\|\widehat{\Sigma} - \Sigma^*\|$, it is inevitable to have lower bounds on the above two quantities.

**Proposition 3.** *Consider the trace regression model* (1) *with* $\varepsilon_i = 0$, $i = 1, \dots, n$. *Then*

$$\operatorname*{argmin}_{\Sigma \in \mathbb{S}_+^m} \frac{1}{2n} \|\mathcal{X}(\Sigma^*) - \mathcal{X}(\Sigma)\|_2^2 = \{\Sigma^*\} \quad \text{for all } \Sigma^* \in \mathbb{T} \cap \mathbb{S}_+^m$$

*if and only if it holds that* $\tau^2(\mathbb{T}) > 0$ *and* $\phi^2(\mathbb{T}) > 0$.

*Correlation constant.* Moreover, we use of the following the quantity. It is not clear to us if it is intrinsically required, or if its appearance in our bound is for merely technical reasons.

$$\mu(\mathbb{T}) = \max \left\{ \frac{1}{n} \langle \mathcal{X}(\Delta), \mathcal{X}(\Delta') \rangle : \ \|\Delta\|_1 \le 1, \Delta \in \mathbb{T}, \ \Delta' \in \mathcal{S}_1^+(m) \cap \mathbb{T}^\perp \right\}.$$

We are now in position to provide a bound on $\|\widehat{\Sigma} - \Sigma^*\|_1$.

**Theorem 2.** *Suppose that model* (6) *holds with* $\Sigma^*$ *as considered throughout this subsection and let* $\lambda_0$ *be defined as in Theorem 1. We then have*

$$\|\widehat{\Sigma} - \Sigma^*\|_1 \le \max \left\{ 8\lambda_0 \frac{\mu(\mathbb{T})}{\tau^2(\mathbb{T})\phi^2(\mathbb{T})} \left( \frac{3}{2} + \frac{\mu(\mathbb{T})}{\phi^2(\mathbb{T})} \right) + 4\lambda_0 \left( \frac{1}{\phi^2(\mathbb{T})} + \frac{1}{\tau^2(\mathbb{T})} \right), \right.$$

$$\left. \frac{8\lambda_0}{\phi^2(\mathbb{T})} \left( 1 + \frac{\mu(\mathbb{T})}{\phi^2(\mathbb{T})} \right), \ \frac{8\lambda_0}{\tau^2(\mathbb{T})} \right\}.$$

**Remark.** Given Theorem 2 an improved bound on the prediction error scaling with $\lambda_0^2$ in place of $\lambda_0$ can be derived, cf. (26) in Appendix D.

The quality of the bound of Theorem 2 depends on how the quantities $\tau^2(\mathbb{T})$, $\phi^2(\mathbb{T})$ and $\mu(\mathbb{T})$ scale with $n$, $m$ and $r$, which is design-dependent. Accordingly, the estimation error in nuclear norm can be non-finite in the worst case and $O(\lambda_0 r)$ in the best case, which matches existing bounds for nuclear norm regularization (cf. Theorem 2 in [19]).

- The quantity $\tau^2(\mathbb{T})$ is specific to the geometry of the constrained least squares problem (7) and hence of critical importance. For instance, it follows from Proposition 1 that for standard Gaussian measurements, $\tau^2(\mathbb{T}) = 0$ with high probability once $n < \delta_m/2$. The situation can be much better for random spd measurements (14) as exemplified for measurements $X_i = z_i z_i^\top$ with $z_i \overset{\text{i.i.d.}}{\sim} N(0, I)$ in Appendix H. Specifically, it turns out that $\tau^2(\mathbb{T}) = \Omega(1/r)$ as long as $n = \Omega(m \cdot r)$.

- It is not restrictive to assume $\phi^2(\mathbb{T})$ is positive. Indeed, without that assumption, even an oracle estimator based on knowledge of the subspace $\mathbb{T}$ would fail. Reasonable sampling operators $\mathcal{X}$ have rank $\min\{n, \delta_m\}$ so that the nullspace of $\mathcal{X}$ only has a trivial intersection with the subspace $\mathbb{T}$ as long as $n \ge \dim(\mathbb{T}) = mr - r(r-1)/2$.

- For fixed $\mathbb{T}$, computing $\mu(\mathbb{T})$ entails solving a biconvex (albeit non-convex) optimization problem in $\Delta \in \mathbb{T}$ and $\Delta' \in \mathcal{S}_1^+(m) \cap \mathbb{T}^\perp$. Block coordinate descent is a practical approach to such optimization problems for which a globally optimal solution is out of reach. In this manner we explore the scaling of $\mu(\mathbb{T})$ numerically as done for $\tau^2(\mathbb{T})$. We find that $\mu(\mathbb{T}) = O(\delta_m/n)$ so that $\mu(\mathbb{T}) = O(1)$ apart from the regime $n/\delta_m \to 0$, without ruling out the possibility of undersampling, i.e. $n < \delta_m$.

## 3 Numerical Results

In this section, we empirically study properties of the estimator $\widehat{\Sigma}$. In particular, its performance relative to regularization-based methods is explored. We also present an application to spiked covariance estimation for the CBCL face image data set and stock prices from NASDAQ.

**Comparison with regularization-based approaches.** We here empirically evaluate $\|\widehat{\Sigma} - \Sigma^*\|_1$ relative to well-known regularization-based methods.

**Setup.** We consider rank-one Wishart measurement matrices $X_i = z_i z_i^\top$, $z_i \overset{\text{i.i.d.}}{\sim} N(0, I)$, $i = 1, \dots, n$, fix $m = 50$ and let $n \in \{0.24, 0.26, \dots, 0.36, 0.4, \dots, 0.56\} \cdot m^2$ and $r \in \{1, 2, \dots, 10\}$ vary. Each configuration of $(n, r)$ is run with 50 replications. In each of these, we generate data

$$y_i = \operatorname{tr}(X_i \Sigma^*) + \sigma \varepsilon_i, \ \sigma = 0.1, \ i = 1, \dots, n, \tag{15}$$

where $\Sigma^*$ is generated randomly as rank $r$ Wishart matrices and the $\{\varepsilon_i\}_{i=1}^n$ are i.i.d. $N(0, 1)$.

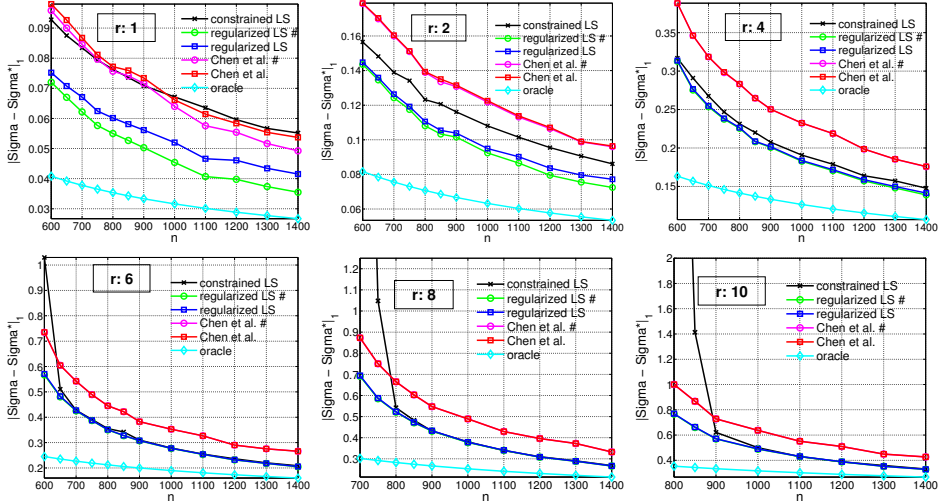

Figure 1: Average estimation error (over 50 replications) in nuclear norm for fixed $m = 50$ and certain choices of $n$ and $r$. In the legend, "LS" is used as a shortcut for "least squares". Chen et al. refers to (16). "#" indicates an oracular choice of the tuning parameter. "oracle" refers to the ideal error $\sigma r \sqrt{m/n}$. Best seen in color.

**Regularization-based approaches.** We compare $\widehat{\Sigma}$ to the corresponding nuclear norm regularized estimator in (11). Regarding the choice of the regularization parameter $\lambda$, we consider the grid $\lambda_* \cdot \{0.01, 0.05, 0.1, 0.3, 0.5, 1, 2, 4, 8, 16\}$, where $\lambda_* = \sigma \sqrt{m/n}$ as recommended in [17] and pick $\lambda$ so that the prediction error on a separate validation data set of size $n$ generated from (15) is minimized. Note that in general, neither $\sigma$ is known nor an extra validation data set is available. Our goal here is to ensure that the regularization parameter is properly tuned. In addition, we consider an oracular choice of $\lambda$ where $\lambda$ is picked from the above grid such that the performance measure of interest (the distance to the target in the nuclear norm) is minimized. We also compare to the constrained nuclear norm minimization approach of [8]:

$$\min_{\Sigma} \operatorname{tr}(\Sigma) \quad \text{subject to} \quad \Sigma \succeq 0, \quad \text{and} \quad \|y - \mathcal{X}(\Sigma)\|_1 \leq \lambda. \tag{16}$$

For $\lambda$, we consider the grid $n\sigma\sqrt{2/\pi} \cdot \{0.2, 0.3, \ldots, 1, 1.25\}$. This specific choice is motivated by the fact that $\mathbf{E}[\|y - \mathcal{X}(\Sigma^*)\|_1] = \mathbf{E}[\|\varepsilon\|_1] = n\sigma\sqrt{2/\pi}$. Apart from that, tuning of $\lambda$ is performed as for the nuclear norm regularized estimator. In addition, we have assessed the performance of the approach in [3], which does not impose an spd constraint but adds another constraint to (16). That additional constraint significantly complicates optimization and yields a second tuning parameter. Thus, instead of doing a 2D-grid search, we use fixed values given in [3] for known $\sigma$. The results are similar or worse than those of (16) (note in particular that positive semidefiniteness is not taken advantage of in [3]) and are hence not reported.

**Discussion of the results.** We conclude from Figure 1 that in most cases, the performance of the constrained least squares estimator does not differ much from that of the regularization-based methods with careful tuning. For larger values of $r$, the constrained least squares estimator seems to require slightly more measurements to achieve competitive performance.

**Real Data Examples.** We now present an application to recovery of spiked covariance matrices which are of the form $\Sigma^* = \sum_{j=1}^r \lambda_j u_j u_j^\top + \sigma^2 I$, where $r \ll m$ and $\lambda_j \gg \sigma^2 > 0$, $j = 1, \ldots, r$. This model appears frequently in connection with principal components analysis (PCA).

**Extension to the spiked case.** So far, we have assumed that $\Sigma^*$ is of low rank, but it is straightforward to extend the proposed approach to the case in which $\Sigma^*$ is spiked as long as $\sigma^2$ is known or an estimate is available. A constrained least squares estimator of $\Sigma^*$ takes the form $\widehat{\Sigma} + \sigma^2 I$, where

$$\widehat{\Sigma} \in \underset{\Sigma \in \mathbb{S}_+^m}{\operatorname{argmin}} \frac{1}{2n}\|y - \mathcal{X}(\Sigma + \sigma^2 I)\|_2^2. \tag{17}$$

The case of $\sigma^2$ unknown or general (unknown) diagonal perturbation is left for future research.

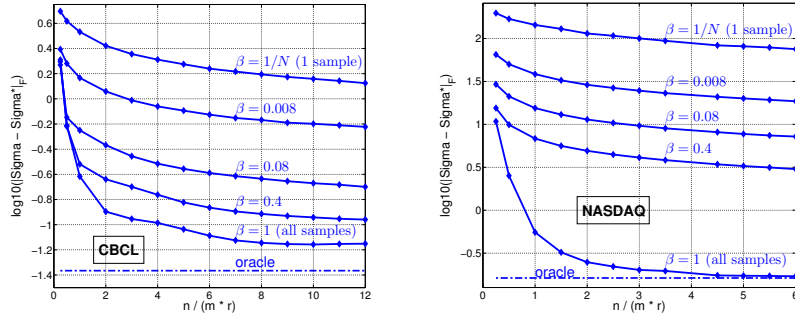

Figure 2: Average reconstruction errors $\log_{10}\|\widehat{\Sigma} - \Sigma^*\|_F$ in dependence of $n/(mr)$ and the parameter $\beta$. "oracle" refers to the best rank $r$-approximation $\Sigma_r$.

**Data sets.** (i) The CBCL facial image data set [1] consist of $N = 2429$ images of $19 \times 19$ pixels (i.e., $m = 361$). We take $\Sigma^*$ as the sample covariance matrix of this data set. It turns out that $\Sigma^*$ can be well approximated by $\Sigma_r$, $r = 50$, where $\Sigma_r$ is the best rank $r$ approximation to $\Sigma^*$ obtained from computing its eigendecomposition and setting to zero all but the top $r$ eigenvalues. (ii) We construct a second data set from the daily end prices of $m = 252$ stocks from the technology sector in NASDAQ, starting from the beginning of the year 2000 to the end of the year 2014 (in total $N = 3773$ days, retrieved from `finance.yahoo.com`). We take $\Sigma^*$ as the resulting sample correlation matrix and choose $r = 100$.

**Experimental setup.** As in preceding measurements, we consider $n$ random Wishart measurements for the operator $\mathcal{X}$, where $n = C(mr)$, where $C$ ranges from 0.25 to 12. Since $\|\Sigma_r - \Sigma^*\|_F / \|\Sigma^*\|_F \approx 10^{-3}$ for both data sets, we work with $\sigma^2 = 0$ in (17) for simplicity. To make recovery of $\Sigma^*$ more difficult, we make the problem noisy by using observations

$$y_i = \text{tr}(X_i S_i), \quad i = 1, \ldots, n, \tag{18}$$

where $S_i$ is an approximation to $\Sigma^*$ obtained from the sample covariance respectively sample correlation matrix of $\beta N$ data points randomly sampled with replacement from the entire data set, $i = 1, \ldots, n$, where $\beta$ ranges from 0.4 to $1/N$ ($S_i$ is computed from a single data point). For each choice of $n$ and $\beta$, the reported results are averages over 20 replications.

**Results.** For the CBCL data set, as shown in Figure 2, $\widehat{\Sigma}$ accurately approximates $\Sigma^*$ once the number of measurements crosses $2mr$. Performance degrades once additional noise is introduced to the problem by using measurements (18). Even under significant perturbations ($\beta = 0.08$), reasonable reconstruction of $\Sigma^*$ remains possible, albeit the number of required measurements increases accordingly. In the extreme case $\beta = 1/N$, the error is still decreasing with $n$, but millions of samples seems to be required to achieve reasonable reconstruction error.

The general picture is similar for the NASDAQ data set, but the difference between using measurements based on the full sample correlation matrix on the one hand and approximations based on random subsampling (18) on the other hand are more pronounced.

## 4 Conclusion

We have investigated trace regression in the situation that the underlying matrix is symmetric positive semidefinite. Under restrictions on the design, constrained least squares enjoys similar statistical properties as methods employing nuclear norm regularization. This may come as a surprise, as regularization is widely regarded as necessary in small sample settings.

## Acknowledgments

The work of Martin Slawski and Ping Li is partially supported by NSF-DMS-1444124, NSF-III-1360971, ONR-N00014-13-1-0764, and AFOSR-FA9550-13-1-0137.

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
