[Supplementary Material]

# A  Proof of Proposition 1

The proof of Proposition 1 follows from results in [2].

**Definition A.1.** *Let $\mathcal{C} \subseteq \mathbb{R}^d$ be a convex cone. The statistical dimension of $\mathcal{C}$ is defined as $\delta(\mathcal{C}) = \mathbf{E}[\|\Pi_{\mathcal{C}}g\|_2^2]$, where $\Pi_{\mathcal{C}}$ denotes the Euclidean projection onto $\mathcal{C}$ and the entries of $g$ are i.i.d. $N(0,1)$.*

**Theorem A.1.** *[2] Let $f : \mathbb{R}^d \to \mathbb{R} \cup \{-\infty, +\infty\}$ be a proper convex function. Suppose that $A \in \mathbb{R}^{n \times d}$ has i.i.d. $N(0,1)$ entries, and let $z_0 = Ax_0$ for a fixed $x_0 \in \mathbb{R}^d$. Consider the convex optimization problem*

$$\text{minimize } f(x) \quad \text{subject to } Ax = z_0. \tag{19}$$

*and let $\mathcal{D}(f, x_0) = \bigcup_{t>0}\{v \in \mathbb{R}^d : f(x_0 + tv) \leq f(x_0)\}$ denote the descent cone of $f$ at $x_0$. Then, for any $\varepsilon > 0$, if $n \leq (1-\varepsilon)\delta(\mathcal{D}(f, x_0))$, with probability at least $1 - 32\exp(-\varepsilon^2 \delta_m)$, $x_0$ fails to be the unique solution of (19).*

*Proof.* (Proposition 1). Define the symmetric vectorization map svec : $\mathbb{S}^m \to \mathbb{R}^{\delta_m}$ by

$$\Sigma = (\sigma_{jk}) \mapsto (\sigma_{11}, \sqrt{2}\sigma_{12}, \ldots, \sqrt{2}\sigma_{1m}, \sigma_{22}, \sqrt{2}\sigma_{23}, \ldots, \sqrt{2}\sigma_{(m-1)m}, \sigma_{mm})^\top, \tag{20}$$

which is an isometry with respect to the Euclidean inner product on $\mathbb{S}^m$ and $\mathbb{R}^{\delta_m}$, and by svec$^{-1}$ : $\mathbb{R}^{\delta_m} \to \mathbb{S}^m$ its inverse. We can then apply Theorem A.1 to the setting of Proposition 1 by using

$$d = \delta_m, \quad x = \text{svec}(\Sigma), \quad x_0 = 0, \quad f(x) = \iota_{\mathbb{S}_+^m}(\text{svec}^{-1}(x)), \quad A = \begin{bmatrix} \text{svec}(X_1) \\ \vdots \\ \text{svec}(X_n) \end{bmatrix},$$

where $\iota_{\mathbb{S}_+^m}$ is the convex indicator function of $\mathbb{S}_+^m$ which takes the value $0$ if its argument is contained in $\mathbb{S}_+^m$ and $+\infty$ otherwise. Observe that $\mathcal{D}(f, 0) = \mathbb{S}_+^m$. It is shown in [2], Proposition 3.2, that the statistical dimension $\delta(\mathbb{S}_+^m) = \delta_m/2$. This concludes the proof. $\square$

# B  Proof of Proposition 2

Proposition 2 follows from the dual problem of the convex optimization problem associated with $\tau^2(\mathcal{X}, R)$. Below, it will be shown that the Lagrangian dual of the optimization problem

$$\min_{A,B} \frac{1}{n^{1/2}}\|\mathcal{X}(A) - \mathcal{X}(B)\|_2$$
$$\text{subject to } A \succeq 0, \ B \succeq 0, \ \text{tr}(A) = R, \ \text{tr}(B) = 1. \tag{21}$$

is given by

$$\max_{\theta, \delta, a} \theta \cdot R - \delta$$
$$\text{subject to } \frac{\mathcal{X}^*(a)}{\sqrt{n}} \succeq \theta I, \qquad \frac{\mathcal{X}^*(a)}{\sqrt{n}} \preceq \delta I, \quad \|a\|_2 \leq 1. \tag{22}$$

The assertion of Proposition 2 follows immediately from (22) by identifying $\theta = \lambda_{\min}(n^{-1/2}\mathcal{X}^*(a))$ and $\delta = \lambda_{\max}(n^{-1/2}\mathcal{X}^*(a))$. In the remainder of the proof, duality of (21) and (22) is established. Using the shortcut $\widetilde{\mathcal{X}} = \mathcal{X}/\sqrt{n}$, the Lagrangian of the dual problem (22) is given by

$$L(\theta, \delta, a; A, B, \kappa) = \theta \cdot R - \delta + \left\langle \widetilde{\mathcal{X}}^*(a) - \theta I, A \right\rangle - \left\langle \widetilde{\mathcal{X}}^*(a) - \delta I, B \right\rangle - \kappa(\|a\|_2^2 - 1).$$

Taking derivatives w.r.t. $\theta, \delta, r$ and the setting the result equal to zero, we obtain from the KKT conditions that a primal-dual optimal pair $(\widehat{\theta}, \widehat{\delta}, \widehat{a}, \widehat{A}, \widehat{B}, \widehat{\kappa})$ obeys

$$\text{tr}(\widehat{A}) = R, \qquad \text{tr}(\widehat{B}) = 1, \qquad \widetilde{\mathcal{X}}(\widehat{A}) - \widetilde{\mathcal{X}}(\widehat{B}) - \widehat{\kappa}2\widehat{a} = 0. \tag{23}$$

Taking the inner product of the rightmost equation with $\widehat{a}$, we obtain

$$\left\langle \widehat{a}, \widetilde{\mathcal{X}}(\widehat{A}) - \widetilde{\mathcal{X}}(\widehat{B}) \right\rangle - \widehat{\kappa}2\|\widehat{a}\|_2^2 = 0.$$

$$\Leftrightarrow \quad \left\langle \widetilde{\mathcal{X}}^*(\widehat{a}), \widehat{A} - \widehat{B} \right\rangle - \widehat{\kappa}2\|\widehat{a}\|_2^2 = 0.$$

$$\Leftrightarrow \quad \widehat{\theta}\operatorname{tr}(\widehat{A}) - \widehat{\delta}\operatorname{tr}(\widehat{B}) - \widehat{\kappa}2\|\widehat{a}\|_2^2 = 0.$$

$$\Leftrightarrow \quad \widehat{\theta}R - \widehat{\delta} = \widehat{\kappa}2\|\widehat{a}\|_2^2,$$

where the second equivalence is by complementary slackness. Consider first the case $\widehat{\theta}R - \widehat{\delta} > 0$. This entails $\widehat{\kappa} > 0$ and thus $\|\widehat{a}\|_2^2 = 1$, so that $2\widehat{\kappa} = \widehat{\theta}R - \widehat{\delta}$. Substituting this result into the rightmost equation in (23) and taking norms, we obtain

$$\widehat{\theta}R - \widehat{\delta} = \|\widetilde{\mathcal{X}}(\widehat{A}) - \widetilde{\mathcal{X}}(\widehat{B})\|_2 = \frac{1}{\sqrt{n}}\|\mathcal{X}(\widehat{A}) - \mathcal{X}(\widehat{B})\|_2. \tag{24}$$

For the second case, note that $\widehat{\theta}R - \widehat{\delta}$ cannot be negative as $a = 0$ is feasible for (22). Thus, $\widehat{\theta}R - \widehat{\delta} = 0$ implies that $\widehat{a} = 0$ and in turn also (24).

## C   Proof of Corollary 1

The corollary follows from Proposition 2 by choosing $a = 1/\sqrt{n}$ so that $n^{-1/2}\mathcal{X}^*(a) = \frac{1}{n}\sum_{i=1}^n X_i$, and using that $\|\Gamma - \widehat{\Gamma}_n\|_\infty \leq \epsilon_n$ implies that $|\lambda_j(\Gamma) - \lambda_j(\widehat{\Gamma}_n)| \leq \epsilon_n$, $j = 1, \ldots, m$ ([12], §4.3). The specific values of $R_*$ and $\tau_*^2$ are obtained by choosing $\zeta = 2$ in Proposition 2.

## D   Proof of Theorem 1

The following lemma is a crucial ingredient in the proof. In the sequel, let $\widehat{\Delta} = \widehat{\Sigma} - \Sigma^*$. Let the eigendecomposition of $\widehat{\Delta}$ be given by

$$\widehat{\Delta} = \sum_{j=1}^m \lambda_j(\widehat{\Delta})u_j u_j^\top = \underbrace{\sum_{j=1}^m \max\{0, \lambda_j(\widehat{\Delta})\}u_j u_j^\top}_{=:\widehat{\Delta}^+} + \underbrace{\sum_{j=1}^m \min\{0, \lambda_j(\widehat{\Delta})\}u_j u_j^\top}_{=:\widehat{\Delta}^-} = \widehat{\Delta}^+ + \widehat{\Delta}^- \tag{25}$$

**Lemma D.1.** *Consider the decomposition* (25). *We have* $\|\widehat{\Delta}^-\|_1 \leq \|\Sigma^*\|_1$.

*Proof.* Write $\widehat{\Delta}^+ = U_+\Lambda_+ U_+^\top$ and $\widehat{\Delta}^- = U_-\Lambda_- U_-^\top$ for the eigendecompositions of $\widehat{\Delta}^+$ and $\widehat{\Delta}^-$, respectively. Since $\widehat{\Sigma} \succeq 0$, we must have $\operatorname{tr}(\widehat{\Sigma}U_-U_-^\top) \geq 0$ and thus

$$\begin{aligned}
0 \leq \operatorname{tr}(\widehat{\Sigma}U_-U_-^\top) &= \operatorname{tr}(U_-^\top\widehat{\Sigma}U_-) \\
&= \operatorname{tr}(U_-^\top(\Sigma^* + \widehat{\Delta})U_-) \\
&= \operatorname{tr}(U_-^\top(\Sigma^* + U_+\Lambda_+U_+^\top + U_-\Lambda_-U_-^\top)U_-) \\
&= \operatorname{tr}(\Sigma^*U_-U_-^\top) + \operatorname{tr}(\Lambda_-),
\end{aligned}$$

where for the last identity, we have used that $U_+^\top U_- = 0$. It follows that

$$\|\widehat{\Delta}^-\|_1 = \|\Lambda_-\|_1 = -\operatorname{tr}(\Lambda_-) \leq \operatorname{tr}(\Sigma^*U_-U_-^\top) \leq \|\Sigma^*\|_1\|U_-U_-^\top\|_\infty = \|\Sigma^*\|_1.$$

$\square$

Equipped with Lemma D.1, we turn to the proof of Theorem 1.

*Proof.* (Theorem 1) By definition of $\widehat{\Sigma}$, we have $\|y - \mathcal{X}(\widehat{\Sigma})\|_2^2 \leq \|y - \mathcal{X}(\Sigma^*)\|_2^2$. Using (6) and the definition of $\widehat{\Delta}$, we obtain after re-arranging terms that

$$\frac{1}{n}\|\mathcal{X}(\widehat{\Delta})\|_2^2 \leq \frac{2}{n}\left\langle \varepsilon, \mathcal{X}(\widehat{\Delta}) \right\rangle = \frac{2}{n}\left\langle \mathcal{X}^*(\varepsilon), \widehat{\Delta} \right\rangle$$

$$\Rightarrow \quad \frac{1}{n}\|\mathcal{X}(\widehat{\Delta})\|_2^2 \leq 2\|\mathcal{X}^*(\varepsilon)/n\|_\infty \|\widehat{\Delta}\|_1 = 2\lambda_0(\|\widehat{\Delta}^+\|_1 + \|\widehat{\Delta}^-\|_1), \tag{26}$$

where we have used Hölder's inequality, the decomposition of $\widehat{\Delta}$ as in Lemma D.1 and $\lambda_0 = \|\mathcal{X}^*(\varepsilon)/n\|_\infty$. We now upper bound the l.h.s. of (26) by invoking Condition 1 and Lemma D.1, which yields $\|\widehat{\Delta}^-\|_1 \leq \|\Sigma^*\|_1$. If $\|\widehat{\Delta}^+\|_1 \leq R_*\|\widehat{\Delta}^-\|_1$, we have

$$\frac{1}{n}\|\mathcal{X}(\widehat{\Sigma}) - \mathcal{X}(\Sigma^*)\|_2^2 = \frac{1}{n}\|\mathcal{X}(\widehat{\Delta})\|_2^2 \leq 2(R_* + 1)\lambda_0\|\Sigma^*\|_1,$$

which is the first part in the maximum of the bound to be established. In the opposite case, suppose first that $\|\widehat{\Delta}^-\|_1 > 0$ (the case $\|\widehat{\Delta}^-\|_1 = 0$ is discussed at the end of this proof) and we have $\|\widehat{\Delta}^+\|_1/\|\widehat{\Delta}^-\|_1 = \widehat{R} > R_* > 1$. Consequently,

$$\frac{1}{n}\|\mathcal{X}(\widehat{\Delta})\|_2^2 = \frac{1}{n}\|\mathcal{X}(\widehat{\Delta}^+) - \mathcal{X}(-\widehat{\Delta}^-)\|_2^2$$

$$= \|\widehat{\Delta}^-\|_1^2 \frac{1}{n}\left\|\mathcal{X}\left(\frac{\widehat{\Delta}^+}{\|\widehat{\Delta}^-\|_1}\right) - \mathcal{X}\left(\frac{-\widehat{\Delta}^-}{\|\widehat{\Delta}^-\|_1}\right)\right\|_2^2$$

$$\geq \|\widehat{\Delta}^-\|_1^2 \min_{\substack{A \in \widehat{R}\mathcal{S}_1^+(m) \\ B \in \mathcal{S}_1^+(m)}} \frac{1}{n}\|\mathcal{X}(A) - \mathcal{X}(B)\|_2^2$$

$$= \tau^2(\mathcal{X}, \widehat{R})\|\widehat{\Delta}^-\|_1^2 = \tau^2(\mathcal{X}, \widehat{R})\frac{\|\widehat{\Delta}^+\|_1^2}{\widehat{R}^2}$$

Inserting this into (26), we obtain the following upper bound on $\|\widehat{\Delta}^+\|_1$.

$$\frac{\tau^2(\mathcal{X}, \widehat{R})}{\widehat{R}^2}\|\Delta^+\|_1^2 \leq 2\lambda_0\frac{\widehat{R} + 1}{\widehat{R}}\|\widehat{\Delta}^+\|_1$$

$$\Rightarrow \quad \|\widehat{\Delta}^+\|_1 \leq 2\lambda_0\frac{\widehat{R}(\widehat{R} + 1)}{\tau^2(\mathcal{X}, \widehat{R})} \leq 4\lambda_0\frac{\widehat{R}^2}{\tau^2(\mathcal{X}, \widehat{R})} \leq 4\lambda_0\frac{R_*^2}{\tau_*^2},$$

where the last inequality follows from the observation that for any $R \geq R_*$

$$\tau^2(\mathcal{X}, R) \geq (R/R_*)^2\tau^2(\mathcal{X}, R_*),$$

which can be easily seen from the dual problem (22) associated with $\tau^2(\mathcal{X}, R)$. Substituting the above bound on $\|\widehat{\Delta}^+\|_1$ into (26) and using the bound $\|\widehat{\Delta}^-\|_1 \leq \|\Sigma^*\|_1$ yields the second part in the maximum of the desired bound. To finish the proof, we still need to address the case $\|\widehat{\Delta}^-\|_1 = 0$. Recalling the definition of the quantity $\tau_0^2(\mathcal{X})$ in (13), we bound

$$\frac{1}{n}\|\widehat{X}(\widehat{\Delta})\|_2^2 = \frac{1}{n}\|\widehat{X}(\widehat{\Delta}^+)\|_2^2 \geq \tau_0^2(\mathcal{X})\|\widehat{\Delta}^+\|_1^2.$$

Inserting this into (26), we obtain from

$$\|\widehat{\Delta}^+\|_1 \leq \frac{2\lambda_0}{\tau_0^2(\mathcal{X})} \leq \frac{2\lambda_0(R_* - 1)^2}{\tau_*^2}, \tag{27}$$

where the second inequality follows from

$$\tau^2(\mathcal{X}, R_*) = \min_{A \in R_*\mathcal{S}_1^+(m) B \in \mathcal{S}_1^+(m)} \frac{1}{n}\|\mathcal{X}(A) - \mathcal{X}(B)\|_2^2$$

$$\leq \min_{A \in \mathcal{S}_1^+(m)} \frac{1}{n}\|\mathcal{X}(R_* \cdot A) - \mathcal{X}(A)\|_2^2 \tag{28}$$

$$= (R_* - 1)^2 \min_{A \in \mathcal{S}_1^+(m)} \frac{1}{n}\|\mathcal{X}(A)\|_2^2 = (R_* - 1)^2\tau_0^2(\mathcal{X})$$

Back-substitution of (27) into (26) yields a bound that is implied by that of Theorem 1. This concludes the proof. □

*Bound on $\lambda_0$.* The bound on $\lambda_0$ is an application of Theorem 4.6.1 in [25].

**Theorem D.1.** *[25] Consider a sequence $\{X_i\}_{i=1}^n$ of fixed matrices in $\mathbb{S}^m$ and let $\{\varepsilon_i\}_{i=1}^n \overset{i.i.d.}{\sim} N(0, \sigma^2)$. Then for all $t \geq 0$*

$$\mathbf{P}\left(\left\|\sum_{i=1}^n \varepsilon_i X_i\right\|_\infty \geq t\right) \leq 2m \exp(-t^2/(2\sigma^2 V^2)), \quad V^2 := \left\|\sum_{i=1}^n X_i^2\right\|_\infty.$$

Choosing $t = \sigma V \sqrt{(1+\mu)2\log(2m)}$ yields the desired bound.

# E  Proof of Theorem 1, Remark 3

The bound hinges on the following concentration result for the extreme eigenvalues of the sample covariance of a Gaussian sample.

**Theorem E.1.** *[9] Let $z_1, \ldots, z_N$ be an i.i.d. sample from $N(0, I_m)$ and let $\Gamma_N = \frac{1}{N}\sum_{i=1}^N z_i z_i^\top$. We then have for any $\delta > 0$*

$$\mathbf{P}\left(\lambda_{\max}\left(\frac{1}{N}\Gamma_N\right) > \left(1 + \delta + \sqrt{\frac{m}{N}}\right)^2\right) \leq \exp(-N\delta^2/2).$$

In the proof, we also make use of the following fact.

**Lemma E.1.** *Let $\{X_i\}_{i=1}^n \subset \mathbb{S}_+^m$. Then*

$$\left\|\sum_{i=1}^n X_i^2\right\|_\infty \leq \max_{1 \leq i \leq n} \|X_i\|_\infty \left\|\sum_{i=1}^n X_i\right\|_\infty.$$

*Proof.* First note that for any $v \in \mathbb{R}^m$ and any $M \in \mathbb{S}_+^m$, we have that

$$v^\top M^2 v = \sum_{j=1}^m \lambda_j^2(M)(u_j^\top v)^2 \leq \lambda_{\max}(M) \sum_{j=1}^m \lambda_j(M)(u_j^\top v)^2 = \|M\|_\infty v^\top X v,$$

where $\{u_j\}_{j=1}^m$ are the eigenvectors of $X$. Accordingly, we have

$$\left\|\sum_{i=1}^n X_i^2\right\|_\infty = \max_{\|v\|_2=1} v^\top \sum_{i=1}^n X_i^2 v \leq \max_{1 \leq i \leq n} \|X_i\|_\infty \max_{\|v\|_2=1} v^\top \sum_{i=1}^n X_i v$$

$$= \max_{1 \leq i \leq n} \|X_i\|_\infty \left\|\sum_{i=1}^n X_i\right\|_\infty.$$

□

We now establish the bound to be shown. Each measurement matrix can be expanded as

$$X_i = \frac{1}{q}\sum_{k=1}^q z_{ik} z_{ik}^\top, \quad \{z_{ik}\}_{k=1}^q \overset{i.i.d.}{\sim} N(0, I_m), \quad i = 1, \ldots, n.$$

Accordingly, we have

$$\left\|\frac{1}{n}\sum_{i=1}^n X_i^2\right\|_\infty = \left\|\frac{1}{n}\sum_{i=1}^n \left\{\frac{1}{q}\sum_{k=1}^q z_{ik} z_{ik}^\top\right\}^2\right\|_\infty$$

$$\leq \max_{1 \leq i \leq n}\left\{\left\|\left\{\frac{1}{q}\sum_{k=1}^q z_{ik} z_{ik}^\top\right\}\right\|_\infty\right\} \left\|\frac{1}{nq}\sum_{i=1}^n\sum_{k=1}^q z_{ik} z_{ik}^\top\right\|_\infty$$

$$\leq \max_{1 \leq i \leq n}\left\{\lambda_{\max}\left(\frac{1}{q}\sum_{k=1}^q z_{ik} z_{ik}^\top\right)\right\} \lambda_{\max}(\Gamma_{nq})$$

where $\Gamma_{nq}$ follows the distribution of $\Gamma_N$ in Theorem E.1 with $N = nq$. For the first term, applying Theorem E.1 with $N = q$ and $\delta = \sqrt{4m \log(n)/q}$ and using the union bound, we obtain that

$$\mathbf{P}\left(\lambda_{\max}\left(\frac{1}{q}\sum_{k=1}^{q} z_{ik}z_{ik}^{\top}\right) > \left(\frac{\sqrt{q} + \sqrt{m} + \sqrt{4m\log n}}{\sqrt{q}}\right)^2\right) \le \exp(-(2m-1)\log n).$$

Applying Theorem E.1 to $\Gamma_N$ with $\delta = 1/\sqrt{q}$, we obtain that

$$\mathbf{P}\left(\lambda_{\max}(\Gamma_{nq}) > \left(1 + \frac{1}{\sqrt{q}} + \sqrt{\frac{m}{nq}}\right)^2\right) \le \exp(-n/2).$$

Combining the two previous bounds yields the assertion.

## F Proof of Proposition 3

In the sequel, we write $\Pi_{\mathbb{T}}$ and $\Pi_{\mathbb{T}^\perp}$ for the orthogonal projections on $\mathbb{T}$ and $\mathbb{T}^\perp$, respectively. Note first that since the $\{\varepsilon_i\}_{i=1}^n$ are zero, any minimizer $\widehat{\Sigma}$ satisfies

$$\mathcal{X}(\widehat{\Sigma}) = \mathcal{X}(\Sigma^*) \iff \mathcal{X}(\widehat{\Delta}) = 0 \iff \mathcal{X}(\widehat{\Delta}_{\mathbb{T}}) + \mathcal{X}(\widehat{\Delta}_{\mathbb{T}^\perp}) = 0 \tag{29}$$

where $\widehat{\Delta}_{\mathbb{T}} = \Pi_{\mathbb{T}}\widehat{\Delta}$ and $\widehat{\Delta}_{\mathbb{T}^\perp} = \Pi_{\mathbb{T}^\perp}\widehat{\Delta}$, where we recall that $\widehat{\Delta} = \widehat{\Sigma} - \Sigma^*$. Note that since $\Sigma^* = \Pi_{\mathbb{T}}\Sigma^*$, for $\widehat{\Sigma}$ to be feasible, it is necessary that $\widehat{\Delta}_{\mathbb{T}^\perp} \succeq 0$.

Suppose first that $\tau^2(\mathbb{T}) = 0$. Then there exist $\Theta \in \mathbb{T}$ and $\Lambda \in \mathcal{S}_1^+(m) \cap \mathbb{T}^\perp$ such that $\mathcal{X}(\Theta) + \mathcal{X}(\Lambda) = 0$. Hence, for any $\Sigma^* \in \mathbb{T}$ with $\Sigma^* + \Theta \succeq 0$, the choices $\widehat{\Delta}_{\mathbb{T}} = \Theta$ and $\widehat{\Delta}_{\mathbb{T}^\perp} = \Lambda$ ensure that $\widehat{\Sigma}$ is feasible and that (29) is satisfied. Since $\Lambda$ is contained in the Schatten 1-norm sphere of radius 1, it is necessarily non-zero and thus $\widehat{\Sigma} \ne \Sigma^*$.
If $\phi^2(\mathbb{T}) = 0$, there exists $0 \ne \Theta \in \mathbb{T}$ such that $\mathcal{X}(\Theta) = 0$. Consequently, for any $\Sigma^* \in \mathbb{T} \cap \mathbb{S}_+^m$ with $\widehat{\Sigma} = \Sigma^* + \Theta \succeq 0$, (29) is satisfied with $\widehat{\Sigma} \ne \Sigma^*$.

Conversely, if $\tau^2(\mathbb{T}) > 0$, (29) cannot be satisfied for $\widehat{\Delta}_{\mathbb{T}^\perp} \succeq 0$, $\widehat{\Delta}_{\mathbb{T}^\perp} \ne 0$. Otherwise, we could divide by $\text{tr}(\widehat{\Delta}_{\mathbb{T}^\perp})$, which would yield

$$\mathcal{X}(\underbrace{\widehat{\Delta}_{\mathbb{T}}/\text{tr}(\widehat{\Delta}_{\mathbb{T}^\perp})}_{\in \mathbb{T}}) + \mathcal{X}(\underbrace{\widehat{\Delta}_{\mathbb{T}^\perp}/\text{tr}(\widehat{\Delta}_{\mathbb{T}^\perp})}_{\in \mathcal{S}_1^+(m) \cap \mathbb{T}^\perp}) = 0,$$

which would imply $\tau^2(\mathbb{T}) = 0$. Therefore, we must have $\widehat{\Delta}_{\mathbb{T}^\perp} = 0$ and $\mathcal{X}(\widehat{\Delta}_{\mathbb{T}}) = 0$, which implies $\widehat{\Delta}_{\mathbb{T}} = 0$ as long as $\phi^2(\mathbb{T}) > 0$.

## G Proof of Theorem 2

Let $\widehat{\Delta} = \widehat{\Sigma} - \Sigma^*$, $\widehat{\Delta}_{\mathbb{T}} = \Pi_{\mathbb{T}}\widehat{\Delta}$ and $\widehat{\Delta}_{\mathbb{T}^\perp} = \Pi_{\mathbb{T}^\perp}\widehat{\Delta} \succeq 0$ as in the preceding proof. We start with the following analog to (26)

$$\frac{1}{n}\|\mathcal{X}(\widehat{\Delta})\|_2^2 = \frac{1}{n}\|\mathcal{X}(\widehat{\Delta}_{\mathbb{T}} + \widehat{\Delta}_{\mathbb{T}^\perp})\|_2^2 \le 2\lambda_0(\|\widehat{\Delta}_{\mathbb{T}}\|_1 + \|\widehat{\Delta}_{\mathbb{T}^\perp}\|_1) \tag{30}$$

Suppose that $\widehat{\Delta}_{\mathbb{T}^\perp} \ne 0$. We then have

$$\|\widehat{\Delta}_{\mathbb{T}^\perp}\|_1^2 \left\{\frac{1}{n}\left\|\mathcal{X}\left(\frac{\widehat{\Delta}_{\mathbb{T}}}{\|\widehat{\Delta}_{\mathbb{T}^\perp}\|_1}\right) + \mathcal{X}\left(\frac{\widehat{\Delta}_{\mathbb{T}^\perp}}{\|\widehat{\Delta}_{\mathbb{T}^\perp}\|_1}\right)\right\|_2^2\right\} \le 2\lambda_0(\|\widehat{\Delta}_{\mathbb{T}}\|_1 + \|\widehat{\Delta}_{\mathbb{T}^\perp}\|_1)$$

Since $\widehat{\Delta}_{\mathbb{T}}/\|\widehat{\Delta}_{\mathbb{T}^\perp}\|_1 \in \mathbb{T}$ and $\widehat{\Delta}_{\mathbb{T}^\perp}/\|\widehat{\Delta}_{\mathbb{T}^\perp}\|_1 = \widehat{\Delta}_{\mathbb{T}^\perp}/\text{tr}(\widehat{\Delta}_{\mathbb{T}^\perp}) \in \mathcal{S}_1^+(m)$, we obtain that the term inside the curly brackets is lower bounded by $\tau^2(\mathbb{T})$ and thus

$$\|\widehat{\Delta}_{\mathbb{T}^\perp}\|_1 \le \frac{2\lambda_0}{\tau^2(\mathbb{T})}\left(1 + \frac{\|\widehat{\Delta}_{\mathbb{T}}\|_1}{\|\widehat{\Delta}_{\mathbb{T}^\perp}\|_1}\right) \tag{31}$$

On the other hand, expanding the quadratic term in (30), we obtain that

$$\frac{1}{n}\|\mathcal{X}(\widehat{\Delta}_{\mathbb{T}})\|_2^2 - \frac{2}{n}\left\langle \mathcal{X}(\widehat{\Delta}_{\mathbb{T}}), \mathcal{X}(\widehat{\Delta}_{\mathbb{T}^\perp})\right\rangle \leq \frac{1}{n}\|\mathcal{X}(\widehat{\Delta})\|_2^2 \leq 2\lambda_0(\|\widehat{\Delta}_{\mathbb{T}}\|_1 + \|\widehat{\Delta}_{\mathbb{T}^\perp}\|_1)$$

$$\Rightarrow \quad \frac{1}{n}\|\mathcal{X}(\widehat{\Delta}_{\mathbb{T}})\|_2^2 \leq 2\lambda_0(\|\widehat{\Delta}_{\mathbb{T}}\|_1 + \|\widehat{\Delta}_{\mathbb{T}^\perp}\|_1) + 2\mu(\mathbb{T})\|\widehat{\Delta}_{\mathbb{T}}\|_1\|\widehat{\Delta}_{\mathbb{T}^\perp}\|_1$$

$$\Rightarrow \quad \phi^2(\mathbb{T})\|\widehat{\Delta}_{\mathbb{T}}\|_1^2 \leq 2\lambda_0(\|\widehat{\Delta}_{\mathbb{T}}\|_1 + \|\widehat{\Delta}_{\mathbb{T}^\perp}\|_1) + 2\mu(\mathbb{T})\|\widehat{\Delta}_{\mathbb{T}}\|_1\|\widehat{\Delta}_{\mathbb{T}^\perp}\|_1$$

$$\Rightarrow \quad \|\widehat{\Delta}_{\mathbb{T}}\|_1 \leq \frac{2\lambda_0\left(1 + \|\widehat{\Delta}_{\mathbb{T}^\perp}\|_1/\|\widehat{\Delta}_{\mathbb{T}}\|_1\right) + 2\mu(\mathbb{T})\|\widehat{\Delta}_{\mathbb{T}^\perp}\|_1}{\phi^2(\mathbb{T})} \tag{32}$$

We now distinguish several cases.

**Case 1**: $\|\widehat{\Delta}_{\mathbb{T}}\|_1 \leq \|\widehat{\Delta}_{\mathbb{T}^\perp}\|_1$. It then immediately follows from (31) that

$$\|\widehat{\Delta}\|_1 \leq \frac{8\lambda_0}{\tau^2(\mathbb{T})} =: T_3. \tag{33}$$

**Case 2a**: $\|\widehat{\Delta}_{\mathbb{T}}\|_1 > \|\widehat{\Delta}_{\mathbb{T}^\perp}\|_1$ and $\|\widehat{\Delta}_{\mathbb{T}^\perp}\|_1 \leq 4\lambda_0/\phi^2(\mathbb{T})$. From (32), we first get

$$\|\widehat{\Delta}_{\mathbb{T}}\|_1 \leq \frac{4\lambda_0 + 2\mu(\mathbb{T})\|\widehat{\Delta}_{\mathbb{T}^\perp}\|_1}{\phi^2(\mathbb{T})} \tag{34}$$

and thus

$$\|\widehat{\Delta}\|_1 \leq \frac{8\lambda_0}{\phi^2(\mathbb{T})}\left(1 + \frac{\mu(\mathbb{T})}{\phi^2(\mathbb{T})}\right) =: T_2 \tag{35}$$

**Case 2b**: $\|\widehat{\Delta}_{\mathbb{T}}\|_1 > \|\widehat{\Delta}_{\mathbb{T}^\perp}\|_1$ and $\|\widehat{\Delta}_{\mathbb{T}^\perp}\|_1 > 4\lambda_0/\phi^2(\mathbb{T})$. Plugging (34) into (31), we obtain that

$$\|\widehat{\Delta}_{\mathbb{T}^\perp}\|_1 \leq \frac{4\lambda_0}{\tau^2(\mathbb{T})} + \frac{4\lambda_0\mu(\mathbb{T})}{\tau^2(\mathbb{T})\phi^2(\mathbb{T})}.$$

Substituting this bound back into (34) yields

$$\|\widehat{\Delta}_{\mathbb{T}}\|_1 \leq \frac{4\lambda_0}{\phi^2(\mathbb{T})} + \frac{8\lambda_0\mu(\mathbb{T})}{\tau^2(\mathbb{T})\phi^2(\mathbb{T})} + \frac{8\lambda_0\mu^2(\mathbb{T})}{\phi^4(\mathbb{T})\tau^2(\mathbb{T})}.$$

Collecting terms, we obtain altogether

$$\|\widehat{\Delta}\|_1 \leq 8\lambda_0\frac{\mu(\mathbb{T})}{\tau^2(\mathbb{T})\phi^2(\mathbb{T})}\left(\frac{3}{2} + \frac{\mu(\mathbb{T})}{\phi^2(\mathbb{T})}\right) + 4\lambda_0\left(\frac{1}{\phi^2(\mathbb{T})} + \frac{1}{\tau^2(\mathbb{T})}\right) =: T_1. \tag{36}$$

Combining (33), (35) and (36) yields the assertion.

## H   Additional Experiments: Scaling of the Constant $\tau^2(\mathbb{T})$

For $\mathcal{X}$ and $\mathbb{T}$ given, it is possible to evaluate $\tau^2(\mathbb{T})$ by solving a convex optimization problem. This is different from other conditions employed in the literature such as restricted strong convexity [17], 1-RIP [8] or restricted uniform boundedness [3] that involve a non-convex optimization problem even for fixed $\mathbb{T}$.

We here consider sampling operators with random i.i.d. measurements $X_i = z_i z_i^\top$, where $z_i \sim N(0, I)$ is a standard Gaussian random vector in $\mathbb{R}^m$ (equivalently, $X_i$ follows a Wishart distribution), $i = 1, \ldots, n$. We expect $\tau^2(\mathbb{T})$ to behave similarly for random rank-one measurements of the same form as long as the underlying probability distribution has finite fourth moments, and thus for (a broad subclass of) the ensemble $\mathcal{M}(\pi_m, q)$ (14).

In order to explore the scaling of $\tau^2(\mathbb{T})$ with $n$, $m$ and $r$, we fix $m \in \{30, 50, 70, 100\}$. For each choice of $m$, we vary $n = \alpha\delta_m$, where a grid of 20 values ranging from 0.16 to 1.1 is considered $\alpha$. For $r$, we consider the grid $\{1, 2, \ldots, m/5\}$. For each combination of $m$, $n$, and $r$, we use 50 replications. Within each replication, the subspace $\mathbb{T}$ is generated randomly from the eigenspace associated with the non-zero eigenvalues of a random matrix $G^\top G$, where the entries of the $m \times r$ matrix $G$ are i.i.d. $N(0, 1)$.

Figure 3: Scaling of $\log \tau^2(\mathbb{T})$ in dependence of $r$ (horizontal axis) and $\alpha = n/\delta_m$ (colors/symbols). The solid lines represent the fit of model (37). Note that the curves are only fitted to those points for which $\tau^2(\mathbb{T})$ exceeds $10^{-6}$. Best seen in color.

The results point to the existence of a phase transition as it is typical for problems related to that under study [2]. Specifically, it turns out that the scaling of $\tau^2(\mathbb{T})$ can be well described by the relation

$$\tau^2(\mathbb{T}) \approx \phi_{m,n} \max\{1/r - \theta_{m,n}, 0\}, \tag{37}$$

where $\phi_{m,n}, \theta_{m,n} > 0$ depend on $m$ and $n$. In order to arrive at model (37), we first obtain the 5%-quantile as summary statistic of the 50 replications associated with each triple $(n, m, r)$. At this point, note that the use of the mean as a summary statistic is not appropriate as it may mask the fact that the majority of the observations are zero. For each pair of $(n, m)$, we then identify all values of $r$ for which the corresponding 5%-quantile drops below $10^{-6}$, which serves as effective zero here. For the remaining values, we fit model (37) using nonlinear least squares (working on a log scale). Figure 3 shows that model (37) provides a rather accurate description of the given data. Concerning $\phi_{m,n}$ and $\theta_{m,n}$, the scalings $\phi_{m,n} = \phi_0 n/m$ and $\theta_{m,n} = \theta_0 m/n$ for constants $\phi_0, \theta_0 > 0$ appear to be reasonable. This gives rise to the requirement $n > \theta_0(mr)$ for exact recovery to be possible in the noiseless case (cf. Proposition 3) and yields that $\tau^2(\mathbb{T}) = \Omega(1/r)$ as long as $n = \Omega(mr)$,

# I Enlarged Figures and Additional Tables

## I.1 Enlarged version of Figure 1

Figure 4: Average estimation error (over 50 replications) in nuclear norm for fixed $m = 50$ and certain choices of $n$ and $r$. In the legend, "LS" is used as a shortcut for "least squares". Chen et al. refers to (16). "#"indicates an oracular choice of the tuning parameter. "oracle" refers to the ideal error $\sigma r \sqrt{m/n}$. Best seen in color.

**Enlarged version of Figure 2**

Figure 5: Average reconstruction errors $\log_{10}\|\widehat{\Sigma} - \Sigma^*\|_F$ in dependence of $n/(mr)$ and the parameter $\beta$. "oracle" refers to the best rank $r$-approximation $\Sigma_r$.

**Additional Tables**

The tables below contain orders of the errors $\|\widehat{\Sigma} - \Sigma^*\|_F$ relative to the error of the best rank $r$ approximation $\|\Sigma_r - \Sigma^*\|_F$ for selected values of $C = n/mr$.

**CBCL**

| $\beta$ | 1 | 1 | .4 | .4 | .08 |
|---|---|---|---|---|---|
| $C$ | 2 | 6 | 4 | 6 | 10 |
| $\frac{\|\widehat{\Sigma}-\Sigma^*\|_F}{\|\Sigma_r-\Sigma^*\|_F}$ | < 3 | < 2 | 4 | 3 | 5 |

**NASDAQ**

| $\beta$ | 1 | 1 | 1 | 1 |
|---|---|---|---|---|
| $C$ | 1 | 2 | 3 | 6 |
| $\frac{\|\widehat{\Sigma}-\Sigma^*\|_F}{\|\Sigma_r-\Sigma^*\|_F}$ | < 3.5 | < 2 | < 1.3 | < 1.1 |

Table 1: Average reconstruction errors relative to $\Sigma_r$ for some selected values of $\beta$ and $C = n/(mr)$.