[Reviews · NeurIPS 2015]

Submitted by Assigned_Reviewer_1

In this paper, the authors propose to study the problem of matrix estimation from linear measurements, where the estimated matrix is assumed to be positive semidefinite (PSD). Under certain constraints on the design, the authors show that ordinary least squares regression, with a PSD constraint, achieves good statistical properties.

More precisely, the authors study the problem of recovering a d x d positive semidefinite (PSD) matrix from n linear measurements. They propose to solve this problem by using an ordinary least squares estimator, and by constraining the estimated matrix to be PSD. Contrary to most previous work, this problem is not regularized.

They start by showing that without any assumptions on the design, the estimation error is unbounded if the number of linear measurements n is too small, even if the matrix to recover has low rank.

They then show that a slow rate on the prediction error can be obtained if the design satisfies a condition introduced in the paper. This condition boils down to having a matrix M in the unit ball of the span of the design (M = sum_i ai Xi, |a| < 1), such that M has its smaller eigenvalue bounded away from zero and its condition number bounded from above. It should be noted that this slow rate coincides with the one for nuclear norm regularized estimators, but it requires an additional condition on the design (while the rate for regularized estimators does not require any conditions).

Finally, the authors introduce a bound on the estimation error, in the case where the matrix to be recovered has low rank r. Three quantities, depending on the design, appear in that bound: the restricted eigenvalue, the separability constant and the correlation constant. My biggest concern is that it is not clear how these two last constants scale with n, d and r. In particular, the authors have not shown how these constants scale for any designs, making it hard to compare this bound with previous work. They have only proposed to study those constants numerically, in the supplementary material. It is thus hard for me to understand how the estimation error behaves, and to know for which designs this estimator would be useful.

I also found the paper hard to read and not particularly well written (e.g. I believe that equations corresponding to lines 267, 272 and 284 should be integrated in sentences. Or the lower blocks in the definition of matrix U line 255 do not make sense).

Overall, I think that this paper could be greatly improved. It seems to me that this work is still a bit too preliminary to be accepted to NIPS.
Summary: In this paper, the authors study the problem of unregularized matrix estimation, where the estimated matrix is positive semidefinite. I believe that this paper is hard to read, and the main bound on the estimation error of low rank matrix depends on quantities which are not studied by the authors (in particular, the scaling of those constants with n, d and r is unknown).

Submitted by Assigned_Reviewer_2

The paper studies the estimation of low-rank semidefinite matrices from noisy inner products with measurement matrices X_1, ..., X_n. Nuclear norm regularized least squares is the dominant approach to estimation in this setting. However, if \Sigma is forced to be positive semidefinite, the nuclear norm reduces to the trace (a single linear function).

The paper proves that under some circumstances, one can accurately estimate \Sigma by simply solving a constrained least squares problem (dropping this linear regularization). This is not possible for general Gaussian measurements, but for measurements that do not have semidefinite matrices in their nullspace (e.g., if the X_i are themselves full rank semidefinite matrices), it is, under conditions.

Simulations verify the claims. The paper also presents a real data experiment, which appears to be somewhat artificial, but again agrees with the theory presented here.

The goal of the paper is to argue that regularization may not be necessary for certain sampling models. How do the theoretical results compare to the best known theoretical results for the regularized problem? The restricted eigenvalue condition is necessary for both formulations; it seems that the separability and correlation conditions are only necessary for the constrained least squares. Comments in the response would be helpful.

Suppose we specialize the results to some random measurement ensemble (e.g., X_i = z_i z_i^*, with z_i a random vector, or X_i a Wishart matrix). Is it possible to evaluate \mu, \tau and \phi, and compare the bound in Theorem 2 to the corresponding result for the nuclear norm regularized variant (11)? Comments in the response would be helpful.

In a similar vein, while it is easy to recognize that the results in this paper are more general than those of [4]/[10], it would be great if we could recover the rates of the special case as a consequence of the general theory introduced here. A comparison would help to understand the strength of the results. Comments in the response would be appreciated.

The main practical implication of the work seems to be that in some situations, it is not necessary to choose a regularization parameter. From a computational perspective, there don't seem to be obvious reasons to prefer (7) to (11).

The paper contains solid results, which seem somewhat under-interpreted. I'm eager to hear the authors responses on the above points, as well as some questions below about the practical motivations for the sampling model.

EDIT: after considering the authors responses on the comparison to nuclear norm minimization and the motivations for the sampling model, I have raised my score to a 6.

Smaller issues:

Can \mu(T) be controlled in terms of \phi(T) and \tau(T)? A similar correlation quantity shows up in the intermediate steps of classical compressed sensing proofs, where it is bounded in terms of restricted isometry constants.

The paper contains a real data experiment showing the estimation of the best rank-r approximation of data covariance, based on inner products with random Wishart matrices. The observations seem to conform (more or less) to the theoretical claims of the paper. However, I'm a little bit puzzled as to the motivation for this experiment. There does not seem to be much practical motivation for this kind of sampling with this kind of data. The message of the paper would be much stronger if it could demonstrate the performance of the proposed method in the context of an application where we really are stuck with a few trace samples.

The results require special measurements which do not have semidefinite matrices in their nullspace (in a quantitative sense). "Generic" gaussian measurements do not satisfy this property, but Wishart measurements do. Besides phase retrieval (where this is a somewhat natural model) are there other practical scenarios where trace sampling with semidefinite X_i applies?
Summary: The paper presents solid, if somewhat under-interpreted results, showing that semidefinite constrained least squares produces accurate estimates of low-rank semidefinite matrices, without any need for trace regularization. The results pertain to certain sampling models -- in particular, to inner products with random semidefinite matrices.

Submitted by Assigned_Reviewer_3

Clarity: The paper is overall well written but can be improved. For example, the structure of Section 3 can be improved. See below. Quality and originality : this paper presents some new progress by generalizing some previous results both on sparse recovery[15,20] and trace regression[4,10].

Detailed comments: (1) It is observed in Figure 1 that when the rank of the underlying matrix is high, more measurements are required for achieving good recovery compared with the regularization based approach. This seems to be a limitation of the proposed method, as in practice the rank may not known a priori and the number of measurements is fixed. In this case, how to predict whether the proposed approach is applicable or not? (2) Eqn.(17) presents extensions to the spiked case with diagonal noise term. However, t sigma^2 is required to be known in advance. In the experiments, sigma^2 = 0 is used. No detailed discussion is provided on its estimation. Another related question is whether it can be generalized beyond diagonal noise case? (3) Section 3 presents several experiments and their associated discussions. Subsection can be used to make the structure of this section better.
Summary: A regualrization free estimation approach is investigated for trace regression with respect to symmetric positive semidefinite (spd) matrices. The finding is that if the underlying matrix is spd and the measurement matrix satisfies certain conditions, simple least square solution may perform as well as trace regularized solution. The paper presents some new results by generalizing some recent progress on sparse recovery and trace regression.

Author Feedback
Author rebuttal: Thanks for your comments and suggestions, and for considering our responses below.

Part 1: General issues

A) Comparison of the bound in Theorem 2 with existing results (Reviewers 1 & 9)

RESPONSE:
i)
The conditions required for Theorem 2 are not directly comparable to those used for the analysis of methods based on nuclear norm regularization. A popular condition is restricted strong convexity (RSC, [16]), which however is not equivalent to our eigenvalue condition in lines 270-272, as for RSC the minimum is taken over a larger set. It is thus not clear whether our conditions imply RSC. A similar reasoning applies to the comparison with the matrix RIP ell1-ell2 condition [8].

ii)
As suggested by Reviewer 1, it makes sense to inspect Theorem 2 for X_i = z_i z_i^T, z_i Gaussian (Wishart measurements). Numerical studies (see supplement for details) indicate the scalings tau(T) = Omega(1/r) and mu(T) = O(1). Moreover, we show that

lambda_0 = O(\sqrt{log(m) log(n) m/n})

Altogether, this yields

|\hat{Sigma} - Sigma^*|_1 = O(r \sqrt{log(m) log(n) m/n} ),

which matches existing bounds for nuclear norm regularization (cf. e.g. Theorem 2 in [18]).

iii)
Our bounds are not comparable to [4, 10] which treat the case r=1 since [4,10] work with bounded noise whereas we assume random Gaussian noise.

B) Practical impact

i)
Our experiments indicate that the performance of constrained least squares (Eq. (7)) is comparable to the corresponding formulation with nuclear norm regularization (11), with slight advantages for the latter as pointed out by Reviewer 5. Moreover, as noted by Reviewer 1, for fixed value of the regularization parameter lambda, (7) and (11) seem to require roughly the same computational effort.

RESPONSE:
While we do agree with these observations, we would like to stress the practical benefits of a regularization-free approach:
The often delicate choice of the regularization parameter is avoided. For our experiments, we ran regularization-based methods under idealized conditions, assuming precise knowledge of the noise model, which makes it straightforward to end up with a close to optimal choice of lambda via a simple grid search and cross-validation. However, this is harder to achieve when the noise level sigma is not known or the noise even deviates from the standard homogeneous sub-Gaussian model. In this case, a coarse-to-fine grid search may be necessary so that the slight additional improvement compared to constrained least squares may no longer be worth the extra computational effort.

ii)
Practical motivations for the sampling model:
As noted by Reviewer 1, the condition for the applicability of our findings is that the sampling operator \cal(X) does not have a positive semidefinite matrix in its nullspace. This is naturally satisfied if the measurement matrices X_i are themselves positive semidefinite. Such positive semidefinite sampling model applies to the following 3 scenarios.

- the quadratic model (5) known from the phase retrieval problem

- low rank or spiked covariance matrix recovery from low-dimensional projections of the data, cf. (4). Parts of our real data experiments are intended to mimic such a scenario as motivated in [8] in the context of high-rate data streams

- standard compressed sensing setup: one is free to choose arbitrary linear measurements

Part 2: Specific issues

REVIEWER 1:
The bound in Theorem 2 depends on quantities phi(T), mu(T) and tau(T).
Can \mu(T) be controlled in terms of phi(T) and tau(T) ? In the RIP-based analysis of compressed sensing, a similar quantity ('restricted orthogonality constant') is bounded in terms of the the RIP constants.

RESPONSE:
This is an interesting connection. As mentioned in the paper, we hypothesize that the dependence on mu(T) is a technical artifact, but we have so far not managed to establish this.

REVIEWER 2:
In the synthetic data experiments, for larger values of the rank r, constrained least squares seems to require more measurements compared to the regularization-based approach. How to predict whether the proposed approach is applicable ?

RESPONSE:
One possible approach might be to evaluate the key quantity \tau with T replaced by \hat{T}, where \hat{T} is defined in correspondence to T, with \Sigma^* replaced by \hat{\Sigma}. Small values of \tau(\hat{T}) can be seen as an indicator to use regularization.

REVIEWER 2:
For the recovery of spiked covariance matrices, the parameter sigma^2 is required to be known. A related issue concerns the case of a non-diagonal noise matrix.

RESPONSE:
This would be interesting extensions to study. Such more general noise models have been discussed in the literature e.g. in 'Latent Variable Graphical Model Selection via Convex Optimization' by Chandasekaran, Parrilo & Willsky (2012). We suggest that we mention this point as a direction of future research in the conclusion.